# Elastic Modulus of a Carbonized Layer on Polyurethane Treated by Ion-Plasma

**DOI:** 10.3390/polym15061442

**Published:** 2023-03-14

**Authors:** Vyacheslav S. Chudinov, Igor N. Shardakov, Yaroslav N. Ivanov, Ilya A. Morozov, Anton Y. Belyaev

**Affiliations:** 1Institute of Continuous Media Mechanics, Ural Branch of Russian Academy of Science, Academician Korolev Street 1, 614013 Perm, Russia; 2Faculty of Mechanics and Mathematics, Perm State University, Bukireva Street 15, 614990 Perm, Russia; 3Department of Computational Mathematics, Mechanics and Biomechanics, Perm National Research Polytechnic University, Komsomolsky Prospect 29, 614990 Perm, Russia

**Keywords:** coatings, mechanical properties, nanomaterials, polyurethane, ion-plasma treatment

## Abstract

Nanocoatings formed by various plasma and chemical methods on the surface of polymeric materials have unique properties. However, the applicability of polymeric materials with nanocoatings under specific temperature and mechanical conditions depends on the physical and mechanical properties of the coating. The determination of Young’s modulus is a task of paramount importance since it is widely used in calculations of the stress–strain state of structural elements and structures in general. Small thicknesses of nanocoatings limit the choice of methods for determining the modulus of elasticity. In this paper, we propose a method for determining the Young’s modulus for a carbonized layer formed on a polyurethane substrate. For its implementation, the results of uniaxial tensile tests were used. This approach made it possible to obtain patterns of change in the Young’s modulus of the carbonized layer depending on the intensity of ion-plasma treatment. These regularities were compared with regularities of changes in the molecular structure of the surface layer caused by plasma treatment of different intensity. The comparison was made on the basis of correlation analysis. Changes in the molecular structure of the coating were determined from the results of infrared Fourier spectroscopy (FTIR) and spectral ellipsometry.

## 1. Introduction

Nanocoatings generated on the surface of polymeric materials by various plasma and chemical methods have unique properties. The surface of nanocoatings differs from the surface of the original polymer material by such properties as wettability, hardness, roughness, permeability, thermal conductivity and electrical conductivity [1,2,3,4,5,6,7,8]. The most promising applications of nanocoatings are flexible nanooptics and nanoplasmonics, flexible electronics and nanosensing [9,10]. Moreover, the developed structure of the material surface, which was formed, for example, after plasma and ion beam treatment, can contain different functional groups, such as graphene and graphite-like nanoclusters with unpaired electrons, providing covalent binding of adsorbed biomolecules [11,12,13].

However, the applicability of polymeric materials with nanocoatings under specific temperature and mechanical conditions depends on the physical and mechanical properties of coatings. The determination of Young’s modulus is the problem of primary concern, as it is widely used in calculations of the stress–strain state of structural elements and entire constructions. The small thickness of the nanocoating makes it difficult to determine the elastic modulus of this layer. There are several approaches to solve this problem. All of them are based on the theoretical interpretation of the experimental results of the response of the “nanocoating-substrate” system to an external mechanical action. The most popular are two approaches [14,15,16,17]. The first one is based on the interpretation of the contact interaction between the cantilever of an atomic force microscope and the nanocoating surface in the framework of the theory of elasticity. The second one is based on the interpretation of the stability of the nanolayer surface within the framework of the theory of elastic stability. It is assumed that the waves on the nanolayer surface are formed during the loss of stability during the deformation interaction of the nanolayer with the substrate. Registration of wavelengths is carried out using atomic force microscopy. The second approach has a well-established name—the strain-induced elastic buckling instability for mechanical measurements (SIEBMM) method. Previously, in [18], it was noted that, if the rigidity of the nanocoating is much higher than the rigidity of the polymer substrate, the elasticity modulus of the coating can be determined by interpreting the results of tests for uniaxial tension of samples with a nanolayer in the framework of an elastic multilayer beam model.

In study [19], an attempt was made to apply the method of uniaxial tensile test for determining the elasticity modulus of metal oxide nanocoatings formed on the Mylar surface (biaxially oriented polyethylene terephthalate) by the atomic layer deposition (ALD) technique. The experiment showed that the influence of the coating on the elastic modulus of the mylar substrate was smaller than the measurement error. That is to be expected, since the modulus of elasticity of the nanolayer material (100–200 GPa) is 25–50 times higher than that of the substrate (4 GPa), whereas the thickness of the substrate is 6000 times greater than that of the layer. The result of the experiment could be predicted using the composite beam theory and the values of the elastic modulus for the mylar material and the metals used, which can be found in the literature.

In this paper we present the results of determination of Young’s modulus for a carbonized layer formed on a polyurethane substrate at different ion-plasma treatment regimes. The basic approach to elastic modulus evaluation is based on the results of uniaxial stretching of polyurethane samples with a carbonized nanolayer. The results obtained using this approach were compared with the results of two approaches described above which used atomic force microscopy (AFM) data. Then, the dependences of the Young’s modulus of the surface layer on the plasma treatment fluence obtained by all three methods were compared with the corresponding changes in the molecular structure of the coating. The comparison was carried out on the basis of the calculation of the correlation coefficients. The molecular structure was estimated from the results of infrared Fourier spectroscopy (FTIR) and spectral ellipsometry. Thus, the achieved degree of carbonization of the surface layer was analyzed depending on the fluence of plasma treatment. It was assumed that the degree of carbonization is the main factor determining the value of the elastic modulus of the modified surface layer.

## 2. Materials and Methods

### 2.1. Preparation of Samples

Polyurethane was synthesized from the urethane prepolymer EP SKU PT-74 based on a simple polyether and 2,4-toluene diisocyanate. Curing of the prepolymer was carried out using a combination of mixtures of hardening agents comprising 3,3′-dichloro-4,4′-diaminodiphenylmethane (mass fraction is 13.2%), polyfuryl (mass fraction is 84.7%) and Voranol RA640 (mass fraction is 2.1%). The polymer and hardener were heated in a thermal chamber at 70 °C in the low vacuum mode. The prepolymer and hardener taken in the ratio of 100:47.5 were mixed for 2 min. Air bubbles were removed from the mixture at 70 °C under low vacuum conditions within 3–4 min. Then, the mixture was cured in three steps: first—for 24 h in silicone mold at 25 °C under low vacuum conditions; second—for 24 h in silicone mold at 50 °C under low vacuum conditions; and third, after removal from the mold, for 3 h under no vacuum conditions at 70 °C. For uniaxial tensile testing, nine rectangular samples of 5 × 40 mm with a thickness of 2 mm were prepared. The AFM scanning of the treated surfaces was performed using four rectangular 5 × 5 mm samples with a thickness of 2 mm.

### 2.2. Ion-Plasma Treatment

In order to form a carbonized nanolayer on the surfaces of polyurethane samples, the latter were treated with nitrogen ions with energy of 20 keV and different fluences of 5 × 10^14^, 10^15^, 5 × 10^15^ ions/cm^2^. Nitrogen ion treatment of the samples was carried out using a specially designed ion-plasma device VSIO-20KV-100NS (Imbiocom Ltd., Perm, Russia). The parameters of treatment were selected to prevent sample overheating. Samples were treated with nitrogen ions in two steps: first, on one side, and then on the other side after turning the sample 180° about the horizontal axis. Such two-sided processing of the sample makes it possible to increase the overall rigidity of the sample during uniaxial deformation due to the formation of two nanolayers with modified properties.

### 2.3. Composite Beam Model

Below are the main relationships for determining the Young’s modulus of a carbonized nanolayer, based on the results of uniaxial tension of flat samples within the elastic model of a compound beam (Figure 1a). The tensile force P in any cross-section of the sample (Figure 1b) is balanced by the stresses σ_s_ and σ_c_ acting, respectively, in the substrate and the layer (coating). Accordingly, the equilibrium equation has the form
(1)$$P=\sigma_{s}\;F_{s}+2\;\sigma_{c}\;F_{c}$$
where F_s_ = b h_s_ and F_c_ = b h_c_ are the cross-sectional areas of the substrate and the layer; σ_s_ and σ_c_ are the stresses in the cross sections of the substrate and layer, respectively.

Let us assume that the substrate and layer materials are linear-elastic. For the polymer substrate, this assumption is valid at small strains and low loading rates. It should be noted that the carbonized layer formed on the surface of polyurethane as a result of ion-plasma treatment has a non-uniform distribution of mechanical properties through thickness of the layer [20]. In view of this fact, the elastic modulus of the layer Ec, obtained in our experimental study should be understood as the effective modulus of elasticity of the carbonized nanolayer. In compliance with Hooke’s law, Equation (1) can be written as
(2)$$P=\varepsilon_{s}\;E_{s}\;F_{s}+2\;\varepsilon_{c}\;E_{c}\;F_{c}$$
where: ε_s_ and ε_c_ are axial deformations, respectively, in the substrate and nanolayer; E_s_ and E_c_ are the Young’s moduli of the substrate and nanolayer materials, respectively; F_s_ and F_c_ are the cross-sectional areas in the sample, respectively, for the substrate and nanolayer.

Using the definition of the effective modulus of elasticity for the whole sample E_eff_, the equilibrium equation can be represented as
(3)$$P=\varepsilon \;E_{\mathrm{eff}}\;F$$
where F = F_s_+ 2 F_c_.

By comparing Equation (2) and Equation (3) considering the compatibility conditions for strains ε = ε_s_ = ε_c_, we arrive at the expression for determining the elastic modulus of the carbonized layer
(4)$$E_{c}=(E_{\mathrm{eff}}h-E_{s}h_{s})/2\;h_{c}$$
where h = h_s_ + 2 h_c_.

### 2.4. Equation for Evaluating the Influence of Carbonized Layer Parameters on the Sample Response to Uniaxial Tension

In the initial untreated substrate, an axial force arising in the cross-section due to a prescribed strain ε is defined as
(5)$$P_{1}=E_{s}\;F\;\varepsilon$$

In the sample treated on both sides, the axial force caused by the deformation ε is determined by the following relation
(6)$$P_{2}=E_{\mathrm{eff}}\;F\;\varepsilon =\left(E_{s}F_{s}+2\;E_{c}F_{c}\right)\varepsilon$$

To assess the measure of the influence of the parameters of the carbonized layer on the reaction of the sample during uniaxial tension, the following relative value was used:(7)$$\delta =(P_{2}-P_{1})/P_{1}*100\%=2\;h_{c}/h\left(E_{c}/E_{s}-1\right)*100\%$$

It follows from this relationship that δ depends on two relative values—the ratio of the elastic moduli of the layer and the substrate (E_c_/E_s_) and the ratio of the thickness of the carbonized layer to the total thickness of the sample (h_c_/h). Thus, the larger the values of h_c_/h and E_c_/E_s_, the more significant the response will be of the treated sample under uniaxial tension compared to the untreated one.

### 2.5. Method for Determining the Elastic Modulus Based on the Results of Uniaxial Tensile Tests

The uniaxial tensile test was performed using a Testometric FS100 (Testometric Company Ltd., Rochdale, United Kingdom) testing machine. The samples were stretched to 1% deformation of the material at a rate of 1%/min. In this deformation mode, the substrate and nanolayer demonstrate a linear-elastic response to external loads.

Based on the results of preliminary studies, it has been found that the strains evaluated from the displacements of the test machine crossbeams and the strains determined from the displacements of the marks on the sample coincide.

Further, the longitudinal elongation was determined only from the displacement of the crossbeam. The resistance force of the sample was measured using the force sensor up to 10 kg.

For one sample, the following sequence of experimental steps was realized:(1)Measurement of linear dimensions of the sample using a Hirox digital optical microscope (Hirox Co Ltd., Tokyo, Japan) and an electronic micrometer.(2)Uniaxial stretching of the sample in a Testometric FS100 testing machine followed by complete unloading. Deformation is carried out up to a value of 1% at a speed of 1% per minute. Registration of the parameters of the dependence “force-deformation”. The loading-unloading cycle is repeated 5 times.(3)Determination of the effective elastic modulus E_eff_ from Relation (3) at each stage of loading and finding the value averaged over 5 loading steps.(4)Ion-plasma treatment of two opposite surfaces of the sample at the specified fluence and energy of ions.(5)Uniaxial stretching of the plasma-treated sample in the mode described in step 2 (5-fold loading-unloading cycle). At each stage of loading-unloading, the modulus of elasticity of the carbonized layer E_c_ is determined by Relation (4). The average value of the module is estimated based on the results of 5 cycles.

### 2.6. Method for Determining Young’s Modulus by Atomic Force Microscopy and the Finite Element Method

The use of atomic force microscopy in the indentation mode is explained by the need to obtain the experimental information and compare it with the results of finite-element analysis. The purpose of such comparison is to determine the modulus of elasticity of the carbonized nanolayer on the surface of polyurethane samples.

The surfaces of the samples with the nanolayer were examined using the atomic force microscope Ntegra Prima (NT-MDT Ltd., Zelenograd, Russia) with the FMG01 (TipsNano) probes with the radius R_p_ of the tip of 10 … 12 nm and stiffness K_p_ of the cantilever was 3.5 N/m. Softer ScanAsyst-Air probes (Bruker) were used for indentation of untreated polyurethanes: R_p_ = 4 nm, K_p_ = 0.4 N/m. The dynamic indentation mode Hybrid 3.0 was realized on the surface of 3 × 3 μm (with 300 × 300 dot resolution). The vertical component of the scanner velocity is equal to a constant value of 10 nm/ms. In this mode, the surface of the nanolayer is indented by the probe tip with high frequency and average force of 27–28 nN. At each point of the examined surface, the interaction of the probe with the surface was recorded as a pressure force versus penetration depth curve. This experimental information was compared with the results of numerical simulation of the contact interaction of the probe with the surface of the carbonized layer on a polyurethane substrate. In the numerical experiment, Young’s modulus of the carbonized layer was determined based on the results of the comparison.

Mathematical modeling was carried out within the framework of the theory of elasticity. The computational scheme for the examined problem in the axisymmetric formulation is shown in Figure 2 in the cylindrical coordinate system (r, z).

For this computational scheme, the following boundary conditions were used: the lower base of the substrate was rigidly clamped (zero value of the displacement vector); the upper horizontal surface of the probe was subjected to a specified constant distributed force, whose resultant was equal to the indentation force produced in the atomic force microscope; the contact boundary between the probe and the carbonized layer surface, which was determined by the iteration procedure, should meet the condition of non-penetration and zero tangential forces; other surfaces remained unloaded.

The carbonized layer and the indenter material were described as isotropic linear elastic materials. Young’s modulus of the indenter material was 200 GPa and the Poisson’s ratio was 0.27. Young’s modulus of the carbonized layer material ranged from 1 to 200 GPa. The Poisson’s ratio of the carbonized layer was set equal to 0.3, which corresponds to carbon materials [21]. The numerical implementation of the mathematical problem of the contact interaction between the probe and the carbonized layer surface was carried out using the finite element method and the ANSYS Mechanical software package.

The static contact problem was solved in the axisymmetric formulation. The axial symmetry was set on side A (Figure 2). Side B was supposed to be rigidly clamped and side C was set free from stresses. The radial dimension of the substrate with the nanolayer was set based on numerical experiments. This dimension was increased until its value no longer influenced the indentation force–penetration depth relationship. The final size of the substrate was 5000 × 2000 nm. The modulus of elasticity of the carbonized layer was found by comparing the experimental data with the numerical results.

Figure 3 shows the image of the finite-element grid near the indenter. Meshing of the carbonized layer is colored green, and meshing of the substrate is colored red.

### 2.7. Determination of Young’s Modulus Using the Results of Atomic Force Microscopy and SIEBMM Method

Relying on the results of atomic force and optical microscopy, it has been established experimentally that the ion-plasma treatment of polyurethane samples with fluences greater than 5 × 10^14^ ions/cm^2^ and ion energy of 20 keV leads to the formation of randomly oriented waves on the sample surface. One could assume that these waves are the result of the elastic instability of the formed carbonized layer under the action of compressive residual stresses. In this case, the experimental information on the wavelength λ can be used to determine the average value of Young’s modulus E_c_ of the carbonized layer in accordance with the strain-induced elastic buckling instability for mechanical measurements (SIEBMM) method [14]. According to this method, Young’s modulus E_c_ is determined from the relation
(8)$$E_{c}={\left(\frac{\lambda }{2\pi h_{c}}\right)}^{3}\cdot \frac{3\left(1-v_{c}^{2}\right)E_{s}}{\left(1-v_{s}^{2}\right)}$$

Surface images of the polyurethane samples after ion-plasma treatment were used to determine the wavelength λ. Imaging of the surface topography was carried out on the area of 20 × 20 μm using a semicontact AFM mode with a resolution of 500 × 500 dots. The range of wavelength variation for each sample was determined based on the mathematical processing of the digital measurements of the surface topography.

### 2.8. Fourier Infrared Spectroscopy and Spectral Ellipsometry

To correlate changes in the elastic modulus of the surface carbon layer obtained by different methods with the changes in the polymer structure due to plasma treatment, the following optical experimental methods were used: Fourier infrared spectroscopy and spectral ellipsometry.

Changes in the chemical structure of the surface layer were analyzed by examining the attenuated total reflection Fourier transform infrared spectra (FTIR ATR) made with a Digilab spectrometer (Australia) at the spectral resolution of 4 cm^−1^ and the number of scans equal to 500. A Harrick FTIR ATR plug-in unit with a 1 × 5 cm^2^ germanium crystal and a beam incidence angle of 45° was used. The spectra were analyzed using Resolution Pro software. To exclude the spectra of water vapor and glue used to fix the germanium crystal in the FTIR ATR plug-in unit, their individual records were taken. The obtained spectra were separately subtracted from the spectrum of polyurethane with normalization coefficients, which were adjusted to each spectrum to ensure complete removal of the water and glue spectra. Note that the depth of infrared beam penetration into the polymer from the germanium crystal varies from 800 to 400 nm depending on the range of the spectrum wave numbers. However, the thickness of the modified surface layer is only 80 nm, which means that the spectrum involves a contribution of the polymer layer lying much deeper than the modified layer. The assessment of changes in the thin surface layer was carried out by subtracting the spectrum of untreated polyurethane from the spectra of the treated polyurethane and selecting the appropriate normalization coefficients. The plots of optical densities of the 1643 cm^−1^ line of unsaturated C=C group vibrations normalized to the optical density of the 1376 cm^−1^ line of the polyurethane macromolecule vibrations were constructed as a function of plasma treatment fluences. These groups correspond to the surface layer carbonization.

The parameters of the carbon nanolayer on the surface of the treated polyurethanes were analyzed using spectral ellipsometry. The spectra were recorded in the light wavelength range of 200–1000 nm using a Woollam M2000 V spectral ellipsometer (J.A. Woollam Company, Lincoln, NE, USA). The number of scans of the spectra was 100. The polarizer pitch was 10 degrees. The ellipsometer was calibrated before measurements using a standard silicone plate with the known thickness of the oxide layer.

After calibration, the spectra of ellipsometric functions psi (Ψ) and delta (Δ) were recorded for untreated polyurethane at different incidence angles of the beam (55, 60, 65, 70 and 75 degrees), as well as for treated polyurethane at the same angles. The theoretical values of the ellipsometric functions Ψ, Δ and values of the refraction index and extinction coefficient were obtained using the Cauchy model—a single-layer model for untreated polyurethane and a two-layer model for treated polyurethane, for which the values of the lower layer parameters were taken from the untreated polymer model [22,23]. Application of the above techniques allowed us to obtain the dependences of the refractive index and extinction coefficients of the surface carbon layer on the fluence of plasma treatment for wavelengths from 200 to 1000 nm.

## 3. Results

The value of the elastic modulus E_s_ of the polyurethane substrate, required for all subsequent calculations, was determined from the results of uniaxial tensile tests. These tests were carried out in accordance with the methodology set out in Section 2.5 of this paper. It was found that the average value of the modulus is E_s_ = 3.1 MPa.

An important parameter for all subsequent evaluations and calculations is the thickness h_c_ of the carbonized layer obtained by ion-plasma treatment of the polyurethane surface with a stream of nitrogen ions. We determined this value based on a theoretical calculation of the process of implantation of nitrogen ions into the surface of the polyurethane substrate. The calculation was carried out using the TRIM [24,25] program. It is known [26] that this program provides a high degree of reliability of the calculation results. We have found that for a flux of nitrogen ions with an energy of 20 keV, the thickness of the carbonized layer is h_c_ = 78 nm.

The possible influence of the carbonized layer on the response of polyurethane specimens to uniaxial tension was estimated using the ratios given in Section 2.4. The total thickness of the specimen was h = 2 mm. The rigidity of the surface layer formed by ion-plasma treatment depends on the type of substrate polymer, as well as on the energy and density of the ion flux, and on the composition of the gas. According to the literature data, the value of the modulus of elasticity of a carbon-like layer can vary from 100 MPa to 100 GPa [27,28]. Thus, the ratio of the modulus of elasticity of the carbonized layer and the substrate (E_c_/E_s_) can be in the range from 32 to 32,000. Using these limiting values and equation (7), we determined the possible interval for the relative increase in axial force δ after double-sided ion-plasma treatment of polyurethane sample-δ ∈ [0,24%, 250%].

Atomic force microscopy was used to plot loading curves at the examined points on the surface of the material. Figure 4 shows the distributions of the depth of penetration into the surface of polyurethanes treated with different doses of ions. The distribution of the penetration depth was obtained at the indentation force of 27–28 nN. The surface of the control sample is heterogeneous because the polyurethane contains soft and hard blocks. As the fluence increases, the surface inhomogeneity decreases, which is manifested as a decrease in the RMS deviation of the penetration depth. The carbonization that takes place in the surface layer of the sample neutralizes the initial difference in mechanical properties between the soft and hard blocks of the surface layer of the material.

The results of numerical calculations by the finite element method were used to plot theoretical curves of the indentation force versus the penetration depth at different moduli of elasticity of the carbon layer (Figure 5). The modulus of elasticity of the carbonized layer was determined by comparing the experimental results of atomic force microscopy with the theoretical data. The points corresponding to the average and extreme values of the penetration depth and the indentation force are plotted on the diagram.

Figure 6 shows the results of uniaxial tensile testing of one of the samples according to the above technique before (solid line) and after (dashed line) plasma treatment. As is seen from the graph, the slope of the curve has changed, which indicates that the elastic properties of the sample have changed after the ion-plasma treatment.

Figure 7 shows the values of the elastic modulus obtained in a series of five uniaxial tension tests for one of untreated samples. Statistical processing of the experimental data showed that the RMS deviation of the elastic modulus from the mean value is less than 1.5%. Such a low spread of values indicates the stability of the elastic modulus and the absence of inelastic deformations.

Figure 8 shows the variation of the relative increase in the effective modulus of the sample ((E_eff_ − E_s_)/E_s_ × 100%) with the fluence before and after ion treatment. It is found that the increase in the effective modulus at all fluences used is greater than three SRM deviations of the values of the elastic modulus obtained from the quintuple measurement.

Figure 9 shows the AFM images of the surface of polyurethane sample after treatment with different fluences.

The values of the elastic modulus of the carbon layer on the surface of the polyurethane sample at different fluences of plasma treatment, which were obtained by the above three methods, are shown in Figure 10.

Figure 11 shows the dependencies obtained by processing the results of infrared spectroscopy and ellipsometry of polyurethane samples treated with plasma having different energy densities. The absorption intensity of the IR spectra was analyzed purposefully for a wave number of 1643 cm^−1^, corresponding to the vibrations of the molecular carbon group C=C. It was found that with increasing fluence, a non-monotonic increase in the absorption intensity of IR spectra by the C=C molecular group occurs. This dependence indicates the carbonization of the polyurethane surface layer and the saturation of this process with an increase in the plasma treatment fluence. Figure 11 shows this dependence in relative units.

The surface layer of polyurethane samples was also analyzed by spectral ellipsometry in accordance with the procedure described in Section 2.8. As a result of processing the measurement data, the dependences of the refractive and extinction indices of the surface carbon layer on the energy flux density of plasma treatment were obtained. These dependences were obtained for wavelengths ranging from 200 nm to 1000 nm. As expected, the carbonization of the surface layer of polyurethane samples affected the refractive and extinction indices. These values show a non-monotonic increase and subsequent saturation with increasing plasma treatment fluence. As follows from the graphs in Figure 11, the nature of these dependences is similar to the change in the parameter obtained by IR spectroscopy.

Thus, the established nature of the dependences in Figure 11 corresponds to the process of carbonization at the molecular level. As is known [11,12,13], after plasma and ion-beam treatment of a polymer material, various functional groups, such as graphene and graphite-like nanoclusters, can form on its surface. First, it is natural to assume that the presence of these functional groups and their concentration will determine the elastic properties of the surface layer. Secondly, the concentration of these groups should be determined by the achieved level of carbonization. A measure of carbonization, in particular, is the intensity of absorption of the IR spectra by the molecular group C=C. From these considerations it follows that the pattern of increase in the elasticity modulus of the surface carbonized polyurethane layer with an increase in the ion treatment fluence should correspond to the pattern of changes in the carbonization process.

In order to establish this correspondence, Pearson’s correlation coefficients [29] were calculated. Each of the three dependencies “modulus of elasticity-fluence” (Figure 10) was compared sequentially with each of the three dependencies characterizing the process of carbonization from fluence (Figure 11). For each pair, the correlation coefficient was calculated. The values of the calculated correlation coefficients are presented in Table 1. Comparison of the values of the coefficients gives reason to believe that the method for determining the modulus of elasticity based on the results of uniaxial tensile tests is preferable compared to the other two.

## 4. Discussion

The values of the elastic modulus determined from the results of uniaxial tensile tests demonstrate statistically significant differences for all fluencies of plasma treatment. An important feature of the proposed method is the use of the same sample for obtaining the elastic modulus of the substrate material and the effective modulus of the sample after ion-plasma treatment. This peculiarity is a necessary condition, because even small differences in the values of elastic modulus between untreated samples may distort the effect of plasma treatment on the experimental results. The condition of formation of a carbon nanolayer on both sides of the sample makes it possible to increase the contribution of the nanolayer material to the strength of the sample.

The results shown in Figure 10 and Figure 11 demonstrate that the values of the elastic modulus of the carbonized layer obtained by uniaxial stretching of the sample are best correlated with the results of IR spectroscopy and ellipsometry as compared to the values determined by other methods. This statement is supported by Table 1, which presents the values of the Pearson correlation coefficient obtained by comparing the results of three methods of determining the elastic modulus of the carbon layer with the data of infrared spectroscopy and ellipsometry.

At the fluencies of plasma treatment of 5 × 10^14^ and 10^15^ ions/cm^2^, the values of elastic modulus calculated by the SIEBMM method (Figure 10) are almost identical, though the optical methods (Figure 11) demonstrate that in this range a significant surface carbonization takes place.

This result can be explained by the fact that the formation of the stress–strain state, determining the loss of stability of the surface layer, is the process of evolutionary nature occurring during ion-plasma treatment. This stress–strain state may not match the conditions justifying the validity of using Equation (8). In our opinion, to establish this correspondence, it is necessary to perform an additional cycle of studies.

It should also be noted that not all material surfaces exhibit a wavy relief after ion-plasma treatment. For example, for stiffer materials with a modulus of elasticity of 150 MPa and more, such as polyethylene, fluoroplastic, polystyrene etc., a wavy surface is not observed and therefore the SIEBMM method is not applicable. In view of the results of this study and the above speculations, it seems reasonable to use SIEBMM and AFM techniques when the elastic properties of the surface nanolayer are comparable to the properties of the substrate or when the ratio of the nanolayer thickness to the substrate thickness is so small that it is impossible to obtain reliable results by applying uniaxial tension.

## 5. Conclusions

In this paper, a methodology is proposed to determine the elastic modulus of the carbonized nanolayer formed by ion-plasma treating the surface of polymeric materials. The technique is based on the results of uniaxial tensile tests.

The applicability of the proposed technique can be predicted based on the knowledge of magnitude of the elastic moduli of the nanolayer and the substrate, as well as the thicknesses of the substrate and layer. In the method for determining the modulus from the results of uniaxial tension, the factors that can affect the values of elastic modulus are reduced to a minimum as compared to other methods.

For the first time it was demonstrated that an experimental method of uniaxial tension can be applied to determine the elastic modulus of the nanocoating on a soft substrate. The values of the elastic modulus obtained by the proposed approach can be used to calculate the stress–strain state of a real product with nanocoating.

## Figures and Tables

**Figure 1 polymers-15-01442-f001:**
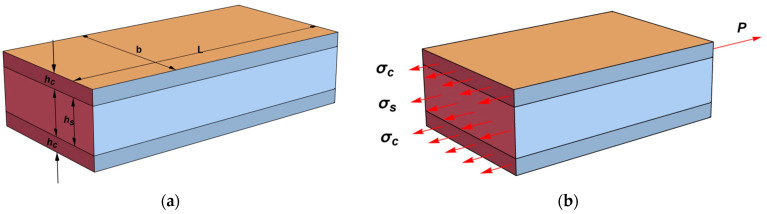
(**a**) Sketch of the sample consisting of a substrate of thickness h_s_ and two surface carbonized layers, each with thickness h_c_. (**b**) Part of the sample in equilibrium under the action of stresses (σ_s_ and σ_c_) and external force (P).

**Figure 2 polymers-15-01442-f002:**
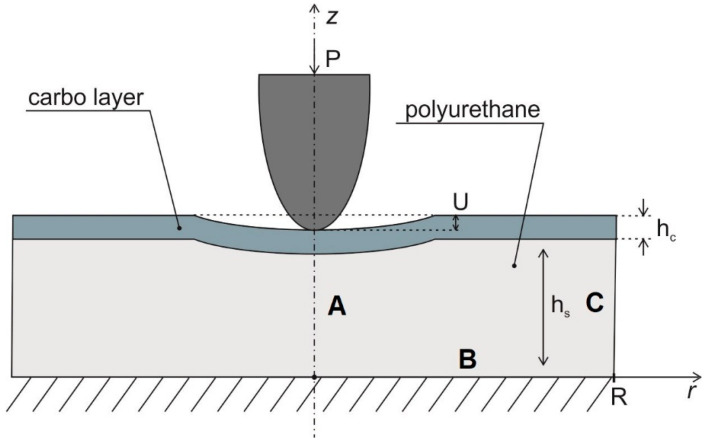
Computational scheme for the mathematical problem of contact interaction of the probe with the carbonized layer surface (h_c_—thickness of carbonized layer; h_s_—thickness of polyurethane base (substrate); P—indentation force; U—depth of probe penetration; R—radius of computational domain).

**Figure 3 polymers-15-01442-f003:**
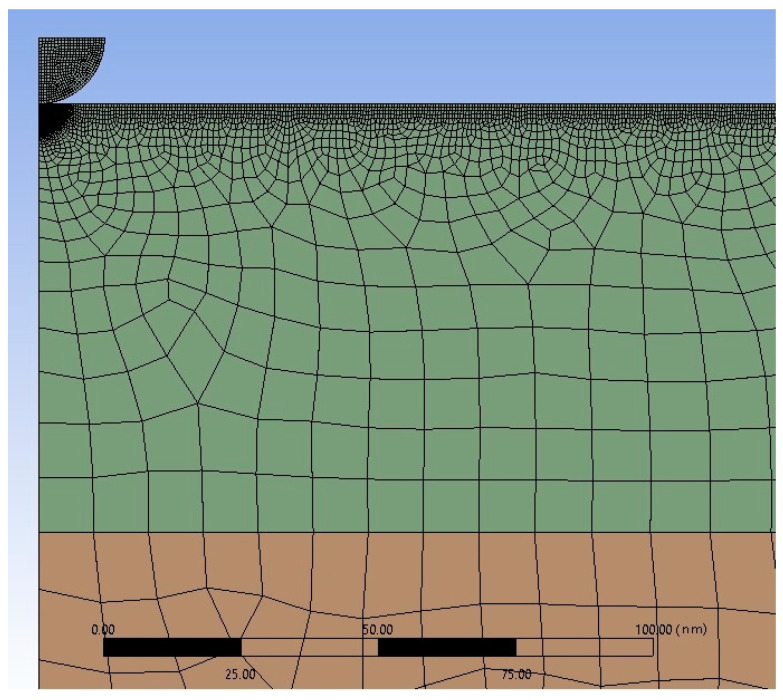
Fragment of the finite-element grid near the indenter. The scale bar is shown at the bottom of the figure.

**Figure 4 polymers-15-01442-f004:**
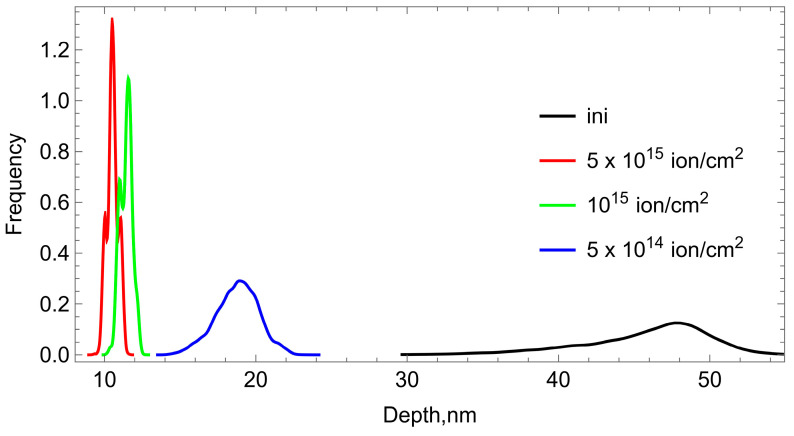
Distribution of maximum penetration depth at the maximum indentation force.

**Figure 5 polymers-15-01442-f005:**
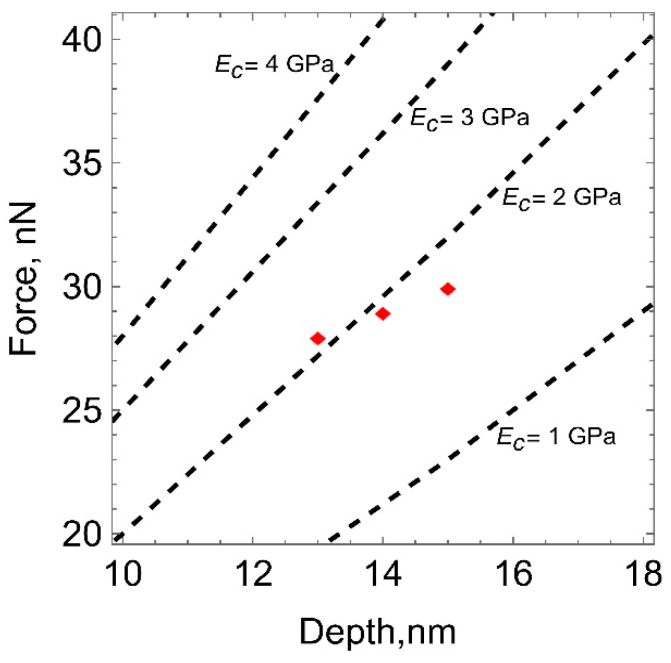
Theoretical curves of the indentation force versus penetration depth at different moduli of layer elasticity (dashed lines). The experimental values corresponding to the indentation of the polyurethane surface after plasma treatment with a fluence of 10^15^ ions/cm^2^ are marked by red dots.

**Figure 6 polymers-15-01442-f006:**
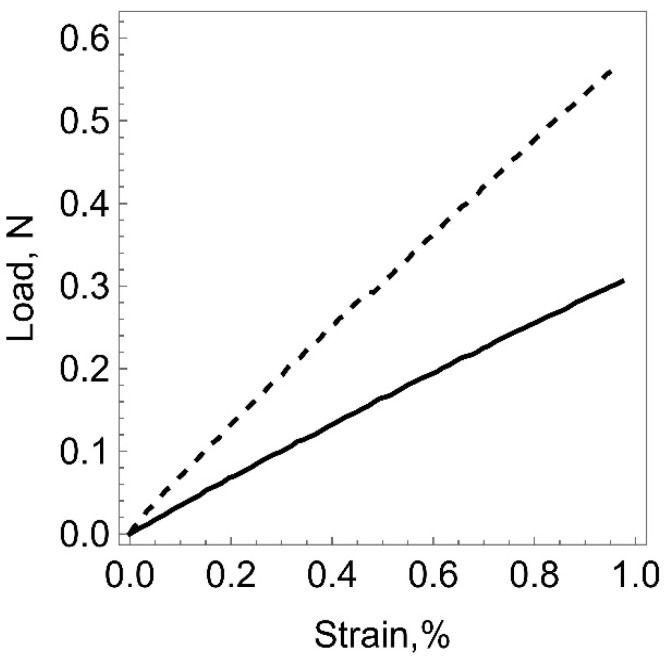
Diagram of polyurethane sample deformation before plasma treatment (solid line) and after treatment with 5 × 10^15^ ion/cm^2^ fluence (dashed line).

**Figure 7 polymers-15-01442-f007:**
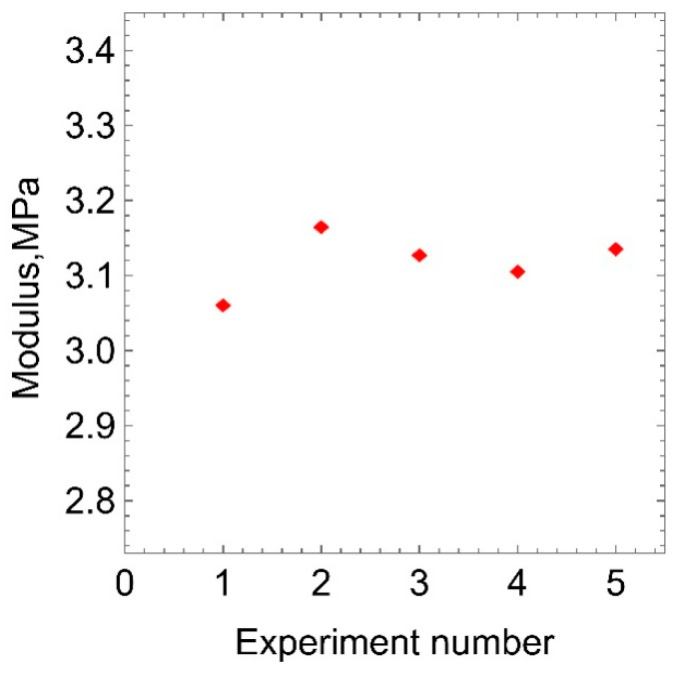
The values of elastic modulus obtained in a series of five experiments for an untreated sample of polyurethane.

**Figure 8 polymers-15-01442-f008:**
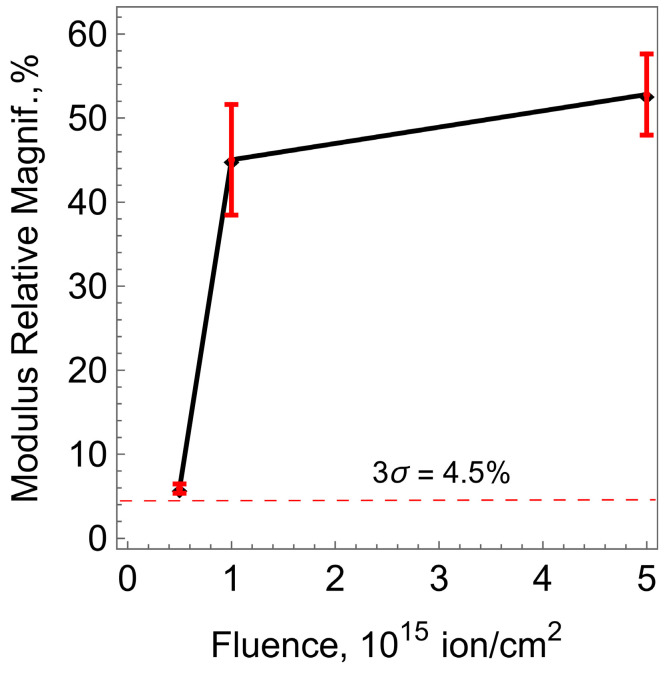
Increase in the effective modulus relative to the initial elastic modulus of the sample as a function of the plasma treatment fluence.

**Figure 9 polymers-15-01442-f009:**
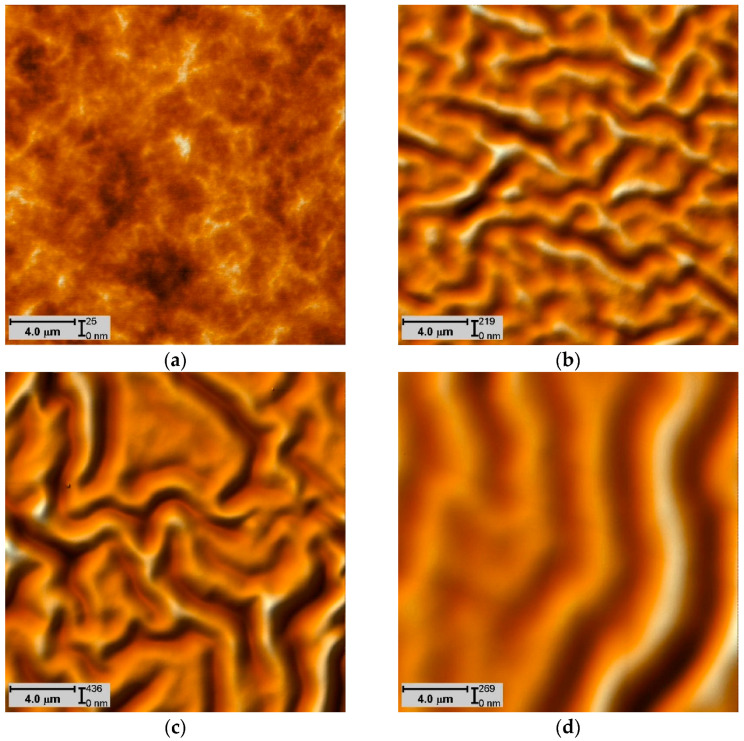
Topography of a 20 × 20 μm region on the surface of untreated polyurethane (**a**) and treated with 5 × 10^14^ ion/cm^2^ (**b**), 10^15^ ion/ cm^2^ (**c**), and 5 × 10^15^ ion/ cm^2^ (**d**).

**Figure 10 polymers-15-01442-f010:**
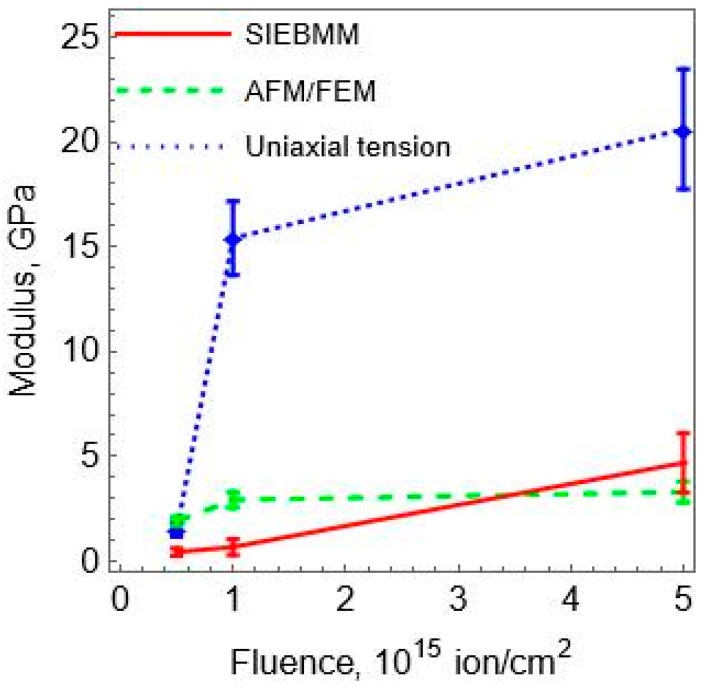
Mean values and RMS deviation of the elastic modulus of the carbonized layer formed at different fluences of ion treatment. The values of the modulus were determined using different techniques: SIEBMM (red), numerical experiment using AFM data (green), uniaxial stretching of the composite beam (blue).

**Figure 11 polymers-15-01442-f011:**
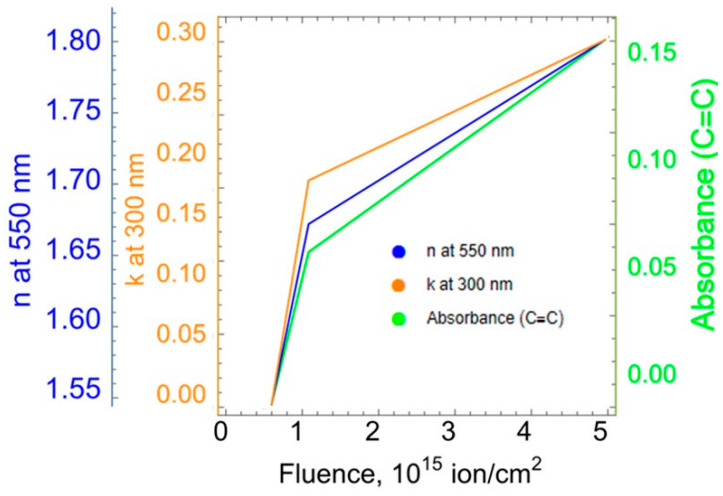
Refraction index at the wavelength of 550 nm (n: blue curve) and extinction coefficient at the wavelength of 300 nm (k: orange curve) of the carbon layer as a function of the fluence of polyurethane plasma treatment; intensity of absorbance (green curve) of lines n(C=C) in the FTIR ATR spectra of polyurethane as a function of the fluence of plasma treatment.

**Table 1 polymers-15-01442-t001:** Values of the Pearson correlation coefficient obtained by comparing the values of elastic modulus obtained by three methods with the values of characteristics indicating carbonization of the surface layer.

	Young’s Modulus Determined by Method Based on the Results of Uniaxial Tensile Tests	Young’s Modulus Determined by Atomic Force Microscopy and the Finite Element Method	Young’s Modulus Determined by the Results of Atomic Force Microscopy and SIEBMM Method
Extinction coefficient at the wavelength of 300 nm	0.99	0.78	0.82
Refraction index at the wavelength of 550 nm	0.97	0.69	0.89
Intensity of absorbance of lines n(C=C) in the FTIR ATR spectra	0.94	0.62	0.93

## Data Availability

The data presented in this study are available on request from the corresponding author.

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
