# Peer review of "Elastic Modulus of a Carbonized Layer on Polyurethane Treated by Ion-Plasma"

_polymers, 2023, doi:10.3390/polym15061442_

Round 1

Reviewer 1 Report

Final result: Major revision.

The manuscript describes the elastic modulus of a carbonized layer on polyurethane treated by ion-plasma. The work is supported by some fruitful measurements and presents useful information. But there are some disadvantages in this manuscript. The authors need to revise them so as to make it acceptable after major revision. My comments are as follows:

1.     Introduction part: The fabrication method of polymeric materials with nanocoatings should be added to the literature. I suggest to add some references in proper place in this section: "Progress in Organic Coatings 157 (2021): 106311; Chemical Physics Letters 755 (2020): 137806. All of these references should be cited to illustrate the fabrication method of polymeric materials with nanocoatings. It is a serious concern.

2.     The abstract should be revised. The key point that you want to share did not illustrate clearly, please rewrite it.

3.     Why not characterize the obtained coating with other material’s mechanical property? Please add the corresponding contents to support your test.

Reviewer 2 Report

The proposed work points the attention of the possibility of use tensile test as tool for the determination of coating elastic properties. Honestly, the paper should be interesting but more than one test should be conducted in order to validate this method. First of all, I’m not in accord with coating choice and Authors do no report any information about its thickness. I could be useful starting from a coating with a “known” tensile behaviour and then make some comparisons. Moreover, Authors say “The application of the experimental method for uniaxial stretching of samples to determine the elastic modulus of the nanocoating on a soft substrate was demonstrated for the first time”. In my opinion this is not true because there is a very important difference between data from AFM indentation and tensile strength and Authors should better justify this conclusion.

In order to improve the paper, in the following a list of corrections:

Abstract must be revised avoiding repetition of concepts

Line 42: please explain SIEBIMM

Line 61: please explain the comparison method and the choice of this comparison

Line 90: please explain what Authors means by “increase the contribution”

Equation 1: probably there is an error in the equation

Line 96: parameters should be defined

Lines 123-138: these are results, not method

Section: are the Authors sure that the 5 fold measurements fall in the elastic region of the stress-strain curves for both bulk and coating?

Line 257: how the measure was performed?

Figure 6: how many samples have been measured? Is this curve an average? Which fold is this?

Line 337: please check the typo error

Figure 10: GPa seems to be too high for carbon based materials

For all of these considerations I’m not sure this paper can be published in Polymers.

All the best

Reviewer 3 Report

The authors presented a method of measuring Young's modulus of a very thin carbon nanolayer on a polyurethane substrate using the uniaxial tensile test.

By varying the conditions for obtaining the nanolayer by plasma treatment, they observed a change in the value of Young's modulus. They compared the obtained data with data obtained using the AFM technique and the finite element method as well as the strain-induced elastic buckling instability for mechanical measurements (SIEBMM) method. The values obtained by different methods differ significantly. Further research is to show that the proposed method gives the most reliable results.

The manuscript is interesting and represents the appropriate level of content, however, it seems to me that the part on the analysis of the results should be extended and justified in more detail. The results presented in Table 1 are not very clear to me. Please describe in more detail the method of calculating the correlation coefficient of the results obtained by the three methods with the parameters of the coatings. I understand that these results are to indicate the credibility of the method presented by the authors, despite slightly different results obtained by other methods. Hence this is a key issue. On the other hand, why is the belief that the parameters obtained by the FTIR ATR and ellipsometry methods must correlate with Young's modulus value of the layer? Please justify it.

Round 2

Reviewer 1 Report

I agree this paper to be published.